

# Application of Levy Processes in Modelling (Geodetic) Time Series With Mixed Spectra

Jean-Philippe Montillet [1,2], Xiaoxing He [3], and Kegen Yu [4]

[1] Institute of Earth Surface Dynamics (IDYST), University of Lausanne, Lausanne, Switzerland
[2] Space and Earth Geodetic Analysis laboratory (SEGAL), University Beira Interior, Covhila, Portugal
[3] School of Civil Engineering and Architecture, East China Jiaotong University, Nan Chang, China
[4] School of Environmental Science and Spatial Informatics, China University of Mining and Technology, Xuzhou, China

**Correspondence:** J.-P. Montillet (jpmontillet@segal.ubi.pt)

**Abstract.** Recently, various models have been developed, including the fractional Brownian motion (fBm), to analyse the stochastic properties of geodetic time series, together with the extraction of geophysical signals. The noise spectrum of these time series is generally modelled as a mixed spectrum, with a sum of white and coloured noise. Here, we are interested in modelling the residual time series, after deterministically subtracting geophysical signals from the observations. This residual

time series is then assumed to be a sum of three random variables (r.v.), with the last r.v. belonging to the family of Levy processes. This stochastic term models the remaining residual signals and other correlated processes. Via simulations and real time series, we identify three classes of Levy processes: Gaussian, fractional and stable. In the first case, residuals are predominantly constituted of short-memory processes. Fractional Levy process can be an alternative model to the fBm in the presence of long-term correlations and self-similarity property. Stable process is characterized by a large variance, which can

be satisfied in the case of heavy-tailed distributions. The application to geodetic time series implies potential anxiety in the functional model selection where missing geophysical information can generate such residual time series.

## 1   Introduction

Among the geodetic data, time series of daily position of Global Navigation Satellite System (GNSS) receiver have been of particular interest for the study of geophysical phenomenon at regional and global scales (e.g., study of the crustal deformation

due to large Earthquakes, sea-level rise -(Bock and Melgare , 2016; Herring et al. , 2016; He et al. , 2017). However, these time series contain white noise and long-memory processes (i.e. coloured noise). The scientific community agrees with the existence of a trade-off in estimating both the stochastic and functional models (He et al. , 2017). More precisely, the choice of the stochastic model directly influences the estimation of the geophysical signals included in the functional model (i.e., tectonic rate, seasonal variations, slow-slip events - (Bock and Melgare , 2016; He et al. , 2017)). To name a few, it includes the

First Order Gauss-Markov (FOGM) model, the white noise with power-law noise (Williams , 2003; Williams et al. , 2004), the Generalized Gauss Markov noise model (GGM), or the Band-pass noise (Langbein , 2008; Langbein and Svarc , 2019). The optimal selection of the stochastic model in GNSS time series analysis remains a hot topic in the scientific community (Bock and Melgare , 2016; Herring et al. , 2016; He et al. , 2017, 2019).





It is widely accepted in the geodesy community (Montillet and Bos , 2019) that most GNSS time series contain flicker
noise which is non-stationary. In addition, recent studies (Langbein and Svarc , 2019; He et al. , 2019) have also advocated the
introduction of a random-walk to model small jumps and residual transient signals which is also a non-stationary stochastic
processes. Thus, several studies, e.g., Montillet and Yu (2015), proposed the use of the fBm, first developed by Mandelbrodt et
al. (1968), in order to model long-memory processes. Botai et al. (2011) and Montillet and Yu (2015) focused on modelling
(residual) geodetic time series using the family of Levy $\alpha$-stable distributions (Nolan , 2009). The application of this family of
distribution was supported by the ability to model long-memory processes and the existence of impulsive signals/noise bursts
in the data sets suggesting deviations from Gaussian distribution (Botai et al. , 2011).

This work discusses several statistical assumptions (i.e. stationary properties, presence of long-term correlations, Gaussian-
ity of the increments) on the underlying processes in the GNSS time series, justifying the application of the fractional Brownian
motion (fBm) and the family of Levy $\alpha$-stable distributions introduced in Montillet and Yu (2015). A significant difference
between Gaussian and Levy stable distributions is that the latter have heavy tails and their variance is infinite. This means that
much larger jumps or flights are possible for Levy stable distributions, which causes their variance to diverge. Many natural
processes follow Levy stable distributions. Therefore this work aims at understanding when the Levy processes can be applied
to model geodetic time series.

The next section starts with the definition of the residual geodetic time series, the fBm and the relationship with the Frac-
tional Autoregressive Integrated Moving Average (FARIMA) model. From financial analysis, we introduce the family of Levy
processes (Panas , 2001) and the assumptions in order to relate to other models (i.e. FARIMA, fBm). Section 3 presents the
assumptions on the use of the Levy processes in the model of the residual time series. To do so, we model the residual geodetic
time series as a sum of three random variables (r.v.), with the hypothesis that the third one is a Levy process. It involves some
justifications compared with established models in the scientific community developed in Section 3.1. In Section 3.2, we de-
velop a $N$ steps method based on the variations of the stochastic and functional models when varying the time series' length.
Section 3.3 is an application to simulated and real time series. Finally, Section 3.4 discusses the limits of modelling geodetic
time series with Levy processes.

## 2   The Stochastic Properties of the Residual Time Series and the Definition of Levy Processes

### 2.1   Model of Residual GNSS Time Series

GNSS time series are generally regarded as a sum of geophysical signals (i.e. seasonal signal, tectonic rate) and stochastic
processes (e.g., white noise, coloured noise) (Williams et al. , 2004; Davis et al. , 2012). Modelling the stochastic processes
within the geodetic time series is crucial in order to estimate the geophysical signal parameters with reliable uncertainties
((Montillet and Bos , 2019) *Chapter* 1 *and* 2, (He et al. , 2017)).

Here, the residual time series are defined as the remaining time series after subtracting deterministically modelled tectonic
rate and seasonal components (i.e. the functional model), from the GNSS observations. The functional model of those signals





is based on the polynomial trigonometric method (Li et al. , 2000; Williams , 2003; Tregoning and Watson , 2009)

$$s_0(t) = at + b + \sum_{j=1}^{N} (c_j \cos(d_j t) + e_j \sin(d_j t)) \tag{1}$$

with $s_0(t)$ the sum of the tectonic rate (with coefficient $a$ and $b$ in Eq. (1)) and the seasonal signal (sum of $cos$ and $sin$ functions in Eq. (1)) at the epoch $t$. Note that $d_j$ is equal to $2\pi j/N$, and $N$ can be equal up to 7 (He et al. , 2017). If $x(t)$ is the residual
time series after subtracting the GNSS time series ($s(t)$) with the functional model ($s_0(t)$) of the geophysical processes (e.g., seasonal signal, tectonic rate), it is generally formulated the hypothesis that the residual time series is a sum of a residual signal and a noise. Following (Williams , 2003; He et al. , 2017; Montillet and Bos , 2019), the stochastic noise model is described with the variance:

$$E\{\mathbf{n}^{\dagger}\mathbf{n}\} = \sigma_{n0}^2 \mathbf{I} + \sigma_{n1}^2 \mathbf{J} \tag{2}$$

where the vector $\mathbf{n} = [n(t_1), n(t_2), ..., n(t_L)]$ is a multivariate noise with $t_i$ the time at the $i$-th epoch. Note that $n(t_i) = n_0(t_i) + n_1(t_i)$, with $n_0(t_i)$ and $n_1(t_i)$ the white noise and the coloured noise sample respectively at the $i$-th epoch. $\dagger$ is the transpose operator, $\mathbf{I}$ the identity matrix, $\mathbf{J}$ is the variance-covariance matrix of the coloured noise. Finally, $\sigma_{n_0}^2$ and $\sigma_{n_1}^2$ are the variance of the white noise and coloured noise respectively. Therefore, this type of time series belongs to the family of mixed spectra, where the mixed spectrum results from the sum of the spectra corresponding to the two kinds of noise (Li , 2013).
Note that the length of the time series $L$ is much larger than the number of frequencies $N$ defining $s_0(t)$.

In the modelling of GNSS time series, a strong assumption is the so-called Gauss-Markov hypothesis ((Montillet and Bos , 2019) *Chapter* 2) which states that the noise is Gaussian distributed and wide sense stationary (WSS). Therefore, we assume the white noise to be zero-mean and Gaussian, whereas the coloured noise with a mean equal to $\mu_C(t)$, slowly varying with time and satisfying the WSS hypothesis (Kasdin , 1995; Haykin , 2002). The distribution of the coloured noise is one of the
key objective of this study, making various assumptions on the type of processes generating this noise.

Finally, the residual signal is considered to be the remaining geophysical signals (i.e. seasonal component and tectonic rate) not completely estimated due to the mismodelling of the stochastic properties of the time series and other small amplitude (i.e. sub millimeter) short time duration transient signals (i.e. local signals, subsidence, ... ) (Bos et al. , 2013; Montillet et al. , 2015; Herring et al. , 2016; He et al. , 2017).

## 2.2 Relationship between the Power-law Noise, fBm and FARIMA

The error spectrum of the GNSS time series is best characterised by a stochastic process following a power-law with index $\beta$. A power-law noise model means that the frequency spectrum is not flat but is governed by long-range dependencies. If the probability density function of the noise is Gaussian or has a different density function with a finite value of variance, its fractal properties can be described by the Hurst parameter ($H$). Montillet et al. (2013) has proposed to use the fractional Brownian
motion (fBm) model in order to model the statistical properties of the residual time series. The essential features of this process are its self-similar behaviour - meaning that magnified and rescaled versions of the process appear statistically identical to the




original - together with its nonstationarity, implying a never-ending growth of variance with time (Mandelbrodt et al. , 1968). It is worth mentioning that a damped version of the fBm exists and known as the Matérn process, defined having a sloped spectrum that matches fBm at high frequencies and taking on a constant value in the vicinity of zero frequency (Lilly et al. ,
90  2017).

Following the definition of the fBm from Mandelbrodt et al. (1968), if $H < 0.5$, the process behaves as a Gaussian variable (anti-persistent); if $H > 0.5$ the process exhibits long-range dependence (persistent); while the case of $H$ equal to 0.5 corresponds to a pure Brownian motion (white noise). Previous studies (Mandelbrodt et al. , 1968; Montillet et al. , 2013) showed that $H$ is directly connected with $\beta$ by the relation:

$$\beta = 2H - 1 \tag{3}$$

With this definition, flicker noise corresponds to $\beta$ equal to 1 or $H$ equal to 1, random walk to $\beta$ equal to 2 or $H$ equal to 1.5, and white noise to $\beta$ equal to 0 ($H$ equal to 0.5). Thus, the random walk and the flicker noise are classified as long-term dependency phenomena (Montillet et al. , 2013). Based on the Hurst exponent, one can favour similar approaches as in financial analysis to deal with modelling stochastic processes.

Long-memory processes are modelled with a specific class of ARIMA models called FARIMA by allowing for non-integer differentiating. A comprehensive literature on the application of FARIMA can be found in financial analysis (Granger and Joyeux , 1980; Panas , 2001). This model can generate long-memory processes based on the value of the different values of the fractional index $d$ (Granger and Joyeux , 1980). When $d$ equal to 0 it is an ARMA process exhibiting short memory; when $-0.5 \leq d < 0$ the FARIMA process is said to exhibit intermediate memory or anti-persistence. This is very similar to the
description of the Hurst parameter in the fBm. There is a relationship between $d$ and $H$ such as $H = d + 0.5$, well-known in financial time series analysis in the presence of aggregation processes (Panas , 2001).

### 2.3  $\alpha$ Stable Random Variable and the Levy $\alpha$-Stable Distributions

In financial analysis, several models are used, including the fBm and the fractional Levy distribution Panas (2001); Wooldridge (2010). The fractional Levy distribution models the Levy processes and in particular the general family of $\alpha$ stable Levy
processes which can be self similar and stationary. Let us recall the definition of a stable random variable.

**Definition** (*Nolan , 2009*), *chap. 1, definition, 1.6* A random variable $X$ is stable if and only if $X \stackrel{d}{=} aZ + b$, where $0 < \alpha \leq 2$, $-1 \leq k \leq 1$, $a \neq 0$, $b \in \mathbb{R}$ and $Z$ is a random variable with characteristic function $\phi(u) = E\{\exp(iuZ)\} = \int_{-\infty}^{\infty} \exp(iuz)F(z) \, dz$. $F(z)$ is the distribution function of $Z$. $E\{.\}$ is the expectation operator. The characteristic function is:

$$\phi(u) = \begin{cases} \exp\left(-|u|^{\alpha}[1 - ik\tan\frac{\pi\alpha}{2}(sign(u))]\right) & if \ \alpha \neq 1 \\ \exp\left(-|u|[1 + ik\frac{2}{\pi}sign(u)]\right), & if \alpha = 1 \end{cases} \tag{4}$$

Where $sign$ is the signum function, $\alpha$ is the characteristic exponent which measures the thickness of the tails of these distributions (the smaller the values of $\alpha$, the thicker the tails of distribution are), $k \in [-1, 1]$ is the symmetry parameter which set the skewness of the distribution. In general, no closed-form expression exists for these distributions, except for the Gaussian ($\alpha$





equal to 2), Pearson ($\alpha$ equal to 0.5, $k$ equal to $-1$) and Cauchy ($\alpha$ equal to 1, $k$ equal to 0) distributions. Note that the distribution is called a symmetric $\alpha$-stable if $k = 0$ (Nolan , 2009; Wang et al. , 2008; Montillet and Yu , 2015). Various methods exist
to estimate the parameters (Koutrouvelis , 1980; Nolan , 2009). In the remainder of this paper, we use the maximum-likelihood method of Nikias and Shao (1995).

Now, if a stochastic process is self-similar, then one can model it with the fBm (see (Cont and Tankov , 2004), Definition 7.1). Following (Weron et al. , 2005), the most commonly used extension of the fBm to the $\alpha$-stable case is the fractional Levy stable motion (fLsm). This process is defined by the integral representation (see appendices). The fLsm is $H$-self-similar and
has stationary increments, with $H$ the Hurst parameter described before. Note that this definition of the Fractional Levy process is different from Benassi et al. (2004) which is not a self-similar process. In the remainder, we use the fLsm definition from ((Weron et al. , 2005), Eq. (6)- recall in the appendices).

Moreover, the relationship between the fLsm and the fBm is obtained from their definition when $\alpha = 2$ (see appendices). If $H = 1/\alpha$, we obtain the Levy $\alpha$-stable motion which is an extension of the Brownian motion to the $\alpha$-stable case. The
Gaussian case (Brownian motion) is then obtained with $\alpha = 2$ (see Weron et al. (2005) for a comprehensive definition of the fLsm). Further definitions such as the fractional stable noise can be established with the fLsm, but there are out of the scope of this work.

Finally, the family of Levy $\alpha$-stable distributions is of a particular interest in this work as the $\alpha$ index is equal to the inverse of the Hurst parameter, therefore in the particular case of the fLsm. Panas (2001) stated that for $1/\alpha < H$, positive increments tend
to be followed by positive increments and long-range dependence (persistence); whereas for $0 < H < 1/\alpha$ positive increments tend to be followed by negative increments (anti-persistence). As a consequence, this family of distributions should be suited when modelling the residual time series with a large amplitude coloured noise with long-memory processes. With the previous definition of the FARIMA and the relationship to $H$, one can assume that the FARIMA model is then favoured over the ARMA process in the case of large coloured noise within the time series. If the white noise is predominant (or $H = 1/2$), the time
series should be fitted with a Gaussian distribution following our assumptions in Section 2.1, and the ARMA model is favoured over the FARIMA.

## 3 Levy Processes Applied to Geodetic Time Series Analysis

This section models the residual GNSS time series as a sum of three r.v. together with the statistical assumptions. We then develop a $N$-steps method to verify our assumptions on simulated and real time series.

### 3.1 Assumptions on the Residual Time Series and the Three Types of Levy Processes

The residual time series is here modelled as a sum of three random variables (r.v.). Namely, it is the sum of a white noise, a coloured noise and a third r.v. It is a similar approach used in previous works looking at the presence of a random-walk component in the stochastic model(Langbein , 2008; Davis et al. , 2012; Langbein and Svarc , 2019; He et al. , 2019). The stochastic properties of the third r.v. should tell us how well is the choice of our initial models (i.e. functional and stochastic).





To recall the definition of the Levy processes in Section 2.3, we postulate that the third r.v. belongs to the Levy processes. We then list the type of Levy processes (Wooldridge , 2010; Cont and Tankov , 2004) depending on the assumptions on the underlying stochastic process:

    1- (Levy Gaussian) The Levy process is a Gaussian Levy process if the r.v. follows the properties of a pure Brownian motion also called a Wiener process (identity variance-covariance matrix, zero-mean, stationary process - (Haykin ,

2002; Wooldridge , 2010)). That is the special case of the fLsm and fBm with $H = 1/2$. The residual time series is assumed to contain mostly short-term correlations modelled with an ARMA process. The residual time series should be modelled with a Gaussian distribution.

    2- (Fractional Levy) The residual time series exhibits self-similarity with possibly long-term correlations. The Fractional Levy process is described by the model of the fLsm for the specific case reduced to the fBm (see previous section). The

long-term correlation process is mostly due to the presence of coloured noise (He et al. , 2017). As explained in Montillet and Yu (2015), the ratio of the amplitude of the coloured over white noise determines which stochastic model of the residual time series should be the most suitable between the FARIMA and ARMA processes. The residual time series should be modelled with a Gaussian distribution following the Gauss-Markov assumption.

    3- (Stable Levy) The Levy process is a Levy $\alpha$-stable motion. That is to generalize important misfit between the selected

(stochastic and functional) model ($s_0(t)$) and the observations. If small jumps (or Markov jumps), outliers or other unknown processes are presents, it results in a distribution of the residual time series potentially (severely) skewed, not symmetric, with possibly heavy tails, hence modelling with a Levy $\alpha$-stable distribution. With the relationship between the Levy $\alpha$-stable motion, the fBm and the FARIMA, we assume that the stochastic properties of the residual time series should be described with the FARIMA, especially in the presence of high amplitude coloured noise.

The assumption of modelling jumps as Markov jumps in the residual GNSS time series may not be intuitive, because the general model is a Heaviside step function (Herring et al. , 2016; He et al. , 2017). Those jumps result from equipment changes (i.e. antenna, radom) to the receiver, sudden events (bumps to the antenna), geophysical nature (co-seismic offsets) and variations in the environment of the receiver occasioning multipath (e.g., growing trees, buildings) (Montillet and Bos , 2019). In financial time series, the jumps are often resulting from the randomness of the stock prices and modelled as random-walk. In addition,

the presence of temporal aggregation processes can affect the persistence in the time series, and sometimes changing suddenly the mean depending on the amplitude of the processes (Working , 1960). That is why in order to assume a Levy $\alpha$-stable motion as the underlying stochastic model in geodetic time series, we restrict our assumption to small undetectable offsets, modelling them potentially as random-walk. For a complete discussion about this topic, we invite readers to refer to Gazeaux et al. (2013) and He et al. (2017).

**3.2 The N Steps Process**

Let us describe the functional model and the stochastic noise model described in Equation (1) and (2) as a functional interpretation called $\mathcal{F}(\theta_1)$ and $\mathcal{G}(\theta_2)$. The functional model described in Equation (1) is then equal in functional form as





$\mathbf{s_0} = [s_0(t_1), s_0(t_2), ..., s_0(t_L)] = \mathcal{F}(\theta_1)$, whereas the stochastic noise model described using the variance-covariance matrix in Equation (2) is equal to $\mathcal{G}(\theta_2)$. We define $\theta_1 = [a, b, (c_j, d_j)_{j=\{1, N\}}]$ and $\theta_2 = [a_{wh}, b_{cl}, \beta]$, the vector parameters for the functional and stochastic noise model respectively. For simplification, we have not included in the functional model the estimation of possible offsets in the time series (see Appendix $B$ for the model). Also, $a_{wh}$ and $b_{cl}$ are the amplitude of the white and coloured noise respectively. The stochastic noise model is here based on the sum of a white and power-law noise ($PL + WN$).

Here, our method is based on varying the length of the time series resulting in the variations of the stochastic and functional models, which they allow classifying the type of Levy process. The variations of the length of the time series should take into account that the coloured noise is a non-stationary signal, and thus the properties (i.e. $b_{cl}$, $\beta$) vary non-linearly. However, varying the length of the time series over several years is not realistic taking into account that real time series can record undetectable transient signals, undocumented offsets and other non-deterministic signals unlikely to be modelled precisely (Montillet et al. , 2015). That is why we restrain the variations of the time series length to 1 year.

Let us call the geodetic time series $\mathbf{s} = [s(t_1), ..., s(t_L)]$ and $\mathbf{s} = [s(t_1), ..., s(t_{L+N})]$ at the first and $N$-th variations respectively. The method can be described as:

$$
\begin{aligned}
\hat{\mathbf{s}} &= \mathcal{F}(\hat{\theta_1}) + \mathcal{G}(\hat{\theta_2}) \; (estimated \; model) \\
1^{st} step : \mathbf{s} &= [s(t_1), ..., s(t_L)] \\
\Delta^1 \mathbf{s} &= \mathbf{s} - [\mathcal{F}(\hat{\theta_1})]_1 \; (residual \; T.S. - \; 1st \; step) \\
&\simeq [\mathcal{G}(\hat{\theta_2})]_1 + \mathbf{res}_1 \\
N^{th} step : \mathbf{s} &= [s(t_1), ..., s(t_{L+N})] \\
\Delta^N \mathbf{s} &= \mathbf{s} - [\mathcal{F}(\hat{\theta_1})]_N \; (residual \; T.S. - \; Nth \; step) \\
&\simeq [\mathcal{G}(\hat{\theta_2})]_N + \mathbf{res}_N
\end{aligned}
\tag{5}
$$

where $\hat{}$ corresponds to the estimated vector or observations. $[.]_j$ means the $j$-th iteration of the estimated quantity. $\Delta^1 \mathbf{s}$ and $\Delta^N \mathbf{s}$ are the residual time series after the first and $N$-th variation of the length of the time series. $\mathbf{res}_1$ and $\mathbf{res}_N$ are the unmodelled signals and stochastic processes after the first and $N$-th step respectively.

To recall the assumptions in Section 3.1, the residual time series $\Delta^N \mathbf{s}$ is modelled as a sum of three r.v. corresponding to the white noise, coloured noise and a Levy process. Using $N$ iterations and the definition of the various Levy processes in the previous section (i.e., Levy Gaussian, Fractional Levy and Stable Levy) in the previous section, we make several assumptions on the estimated parameters and selected stochastic models in order to characterize this third r.v. Table 1 summarises the assumptions for these three cases. We use specific mathematical symbols to differentiate between them. $\triangleq$ means the equality in terms of distribution. $\simeq$, $\sim$ and $\neq$ are related to the variations of the estimated parameters of the stochastic model associated with the first and the $N$-th iteration. This variation is calculated using the sum of the difference in absolute value between the parameters between the first and the $N$-th iteration. Then, a percentage is deduced based on the initial value of the parameters (at first iteration). Now specifically, the symbol $\simeq$ means that there are little differences (less than $3\%$) between the estimated





parameters of the stochastic model associated with the first and the $N$-th iteration. The symbol $\sim$ means that we allow bigger differences up to $20\%$. With much larger values, we use the symbol $\neq$.

Moreover, the estimation of the model parameters is carried out using the Hector software (Bos et al. , 2013). We have restrained our processing to the stochastic model corresponding to the flicker noise (with white noise - $FN + WN$) and power-

law (with white noise $PL + WN$). The optimal choice of the stochastic model is a current research topic in GNSS time series analysis including recent studies such as He et al. (2017), He et al. (2019) and Montillet and Bos (2019). To simplify our study, we have preliminarily applied the method based on the Akaike information criterion developed in He et al. (2019) on the real time series to select the stochastic noise model. Therefore we have selected real time series with stochastic models $FN + WN$ and $PL + WN$. We are not going to develop further this selection process in this study, but readers can refer to

He et al. (2019).

Furthermore, the fitting of the ARMA($p$,$q$) and FARIMA($p$,$d$,$q$) model to the residual time series is carried out by maximum likelihood following Sowell (1991), varying the lags $p$ and $q$ within the interval $[0,5]$. Note that the fractional parameter $d$ is an output of the software Hector (Bos et al. , 2013) when fitting the stochastic model during the $N$ iterations. Also, the ARMA/FARIMA model which best fits the residual time series, is selected in order to minimize the Bayesian Information

Criterion (BIC) following Montillet and Yu (2015). Finally, one can wonder if the anxiety in the model selection (ARMA, FARIMA) in presence of heavy-tails can modify the performance of the BIC. This topic is currently debated in the statistical community (see Panahi (2016)). Large tails should be detected in the fitting of the Levy $\alpha$-stable distribution via the maximum-likelihood method of Nikias and Shao (1995). Due to the direct relationship between the index $\alpha$ and $H$, we assume that the FARIMA should be chosen defacto over the ARMA model.

**Table 1.** Assumptions on the functional model and the stochastic parameters estimated via $N$ iterations (see,$N$-Step method) to characterize the type of Levy processes within the geodetic time series. The symbols and notations are explained in Section 3.2

| *Type of Process* | Levy Gaussian | Fractional Levy | Stable Levy |
|---|---|---|---|
| *Mathematical* | $[\mathcal{G}(\hat{\theta_2})]_1 \simeq [\mathcal{G}(\hat{\theta_2})]_N$ | $[\mathcal{G}(\hat{\theta_2})]_1 \sim [\mathcal{G}(\hat{\theta_2})]_N$ | $[\mathcal{G}(\hat{\theta_2})]_1 \neq [\mathcal{G}(\hat{\theta_2})]_N$ |
| *Assumptions* | $[\mathcal{F}(\hat{\theta_1})]_1 \simeq [\mathcal{F}(\hat{\theta_1})]_N$ | $[\mathcal{F}(\hat{\theta_1})]_1 \sim [\mathcal{F}(\hat{\theta_1})]_N$ | $[\mathcal{F}(\hat{\theta_1})]_1 \neq [\mathcal{F}(\hat{\theta_1})]_N$ |
| *(Distribution)* $\Delta^1 \mathbf{s} \triangleq$ | Gaussian | Gaussian | Levy $\alpha$-stable |
| *Model To Characterize Processes* | ARMA(p,q) | ARMA(p,q) or FARIMA(p,d,q) | FARIMA(p,d,q) |

### 3.3  Application to Simulated and Real Time Series

#### 3.3.1  Simulated Time Series

The definition of the Levy processes together with the assumptions in Table 1 are applied to the residual of simulated geodetic time series. The simulations of the geodetic time series follow Williams et al. (2004) and the routines associated with Hector (Bos et al. , 2013). The estimations of the ARMA and FARIMA models follow Section 3.2.


**Figure 1.** Percentage of variations of the estimated parameters included in the stochastic and functional models when varying the length of the time series. $(A)$, $(B)$ and $(C)$ refer to the various scenarios with different coloured noise amplitude.

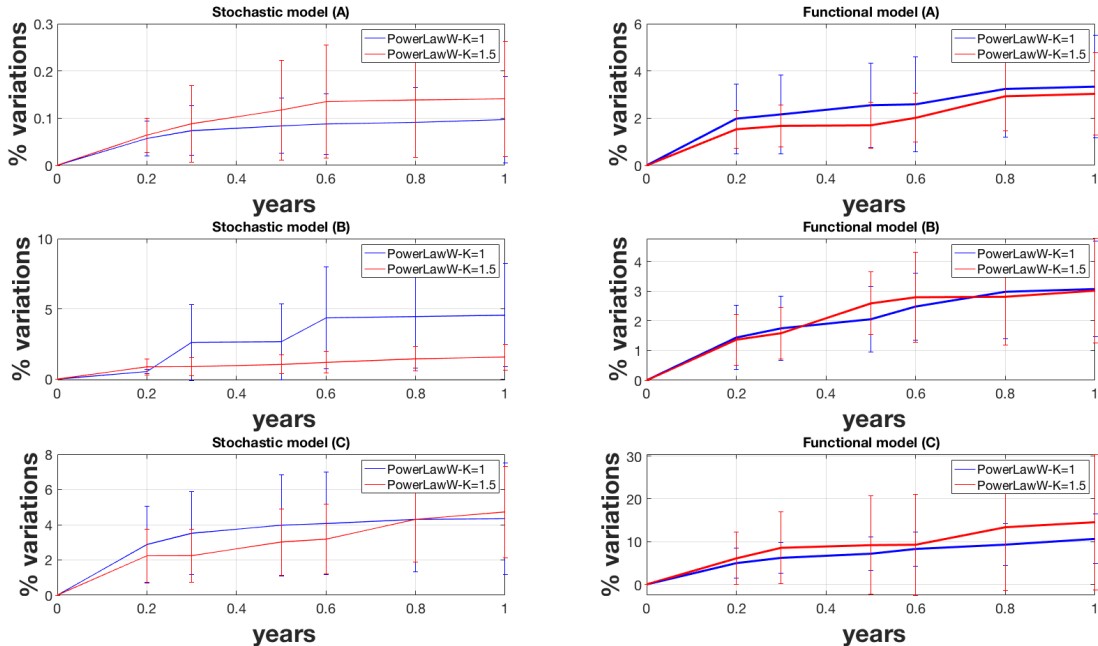

We simulate 10 years long time series fixing $a_{wh}$ to 1.6 mm, $a$ varying between $[1-3]$ mm/yr, $b$ equal 0, and $(c_1, d_1)$ equal to $(0.4, 0.2)$ mm/yr. According to Table 1, we vary the amplitude of coloured noise $b_{cl}$ following three scenarios: $(A)$ from low value (i.e. $b_{cl} < 0.1$ mm/$yr^{\beta/4}$); $(B)$ intermediate (i.e. $1mm/yr^{\beta/4} > b_{cl} > 0.1$ mm/$yr^{\beta/4}$); and $(C)$ high value (i.e. $1mm/yr^{\beta/4} < b_{cl} < 4mm/yr^{\beta/4}$). In the case of the large amplitude of the coloured noise, the process is unlikely zero-mean stationary. Also, $\beta$ is equal to 1 (flicker noise) or 1.5 (power-law noise) in the simulations.

Figure 1a, 1b and 1c display the results when averaging over 50 time series. The variations are in steps of $[0, 0.3, 0.5, 0.7, 0.8, 1]$ year (see X-axis). Each figure corresponds to the different coloured noise amplitude following the three scenarios described above. We show both the variations of the stochastic and functional models. On the Y-axis, these variations are basically the statistics (mean and standard deviation) over the percentage estimated between the parameters of either the stochastic or functional model between the first and $N$-th iteration. For each run, this percentage is calculated using the sum of the differences in absolute value of the various parameters described in Section 3.2.

In Hector, we use the $PL + WN$ model (Bos et al. , 2013). The first result which is common to all three figures, is that the variations in the functional model starts earlier than for the stochastic model. Previous studies have shown that there is some part of the noise amplitude absorbed in the functional model (Williams , 2003; Montillet et al. , 2015). In our scenario, the estimation of the linear trend may fit partially into the power-law noise, hence reducing the variations of the stochastic model.





This effect can be amplified with higher spectral indexes. Now, Figure 1 shows that over 1 year the variations of the stochastic and functional model are less than $4\%$ (mean value) for small amplitude coloured noise, whereas when increasing the coloured noise the variations increase quickly (e.g., more than $20\%$ for the large coloured noise amplitude for the functional model $(c)$). Knowing that Hector assumes only stationary signals (Bos et al. , 2013), it means that part of the large variations of the coloured noise is wrongly included in the estimation of the functional model.

**Table 2.** Statistics on the Error when fitting the ARMA and FARIMA model to the residual time series following the three scenarios

| *Error (mm)* | | case $A$ | case $B$ | case $C$ |
|---|---|---|---|---|
| | $\beta$ | $b_{cl} < 0.1$ mm/$yr^{\beta/4}$ | $1mm/yr^{\beta/4} > b_{cl} > 0.1$ mm/$yr^{\beta/4}$ | $1mm/yr^{\beta/4} < b_{cl} < 3mm/yr^{\beta/4}$ |
| *ARMA* | 1.1 | $1.44 \pm 0.01$ | $1.74 \pm 0.01$ | $1.89 \pm 0.04$ |
| | 1.5 | $1.46 \pm 0.01$ | $1.76 \pm 0.04$ | $1.95 \pm 0.05$ |
| *FARIMA* | 1.1 | $1.91 \pm 0.02$ | $1.85 \pm 0.02$ | $1.46 \pm 0.02$ |
| | 1.5 | $1.89 \pm 0.01$ | $1.75 \pm 0.03$ | $1.59 \pm 0.05$ |

**Table 3.** Correlation between the distribution of the residuals and the Normal ($Corr.\ Normal$) and the Levy $\alpha$-stable distribution ($Corr.\ Levy$) following the three scenarios

| *Corr.* $[0-1]$ | | case $A$ | case $B$ | case $C$ |
|---|---|---|---|---|
| | $\beta$ | $b_{cl} < 0.1$ mm/$yr^{\beta/4}$ | $1mm/yr^{\beta/4} > b_{cl} > 0.1$ mm/$yr^{\beta/4}$ | $1mm/yr^{\beta/4} < b_{cl} < 3mm/yr^{\beta/4}$ |
| *Corr. Normal* | 1.1 | $0.93 \pm 0.14$ | $0.92 \pm 0.21$ | $0.89 \pm 0.50$ |
| | 1.5 | $0.92 \pm 0.14$ | $0.91 \pm 0.22$ | $0.85 \pm 0.31$ |
| *Corr. Levy* | 1.1 | $0.92 \pm 0.11$ | $0.94 \pm 0.14$ | $0.96 \pm 0.18$ |
| | 1.5 | $0.93 \pm 0.13$ | $0.94 \pm 0.16$ | $0.95 \pm 0.18$ |

Now, Table 2 shows the standard deviation of the difference ($Mean\ Square\ Error$) between the ARMA /FARIMA model and the residuals (i.e. $\mathbf{res}_i$ in Equation (5)). We do not display any mean, because the fit of the models are done on the zero-mean residuals. Note that the value is averaged over the 50 simulations, together with the variations of the length of the time series following the same processing as before. The table also displays the averaged correlation between the distribution of the residuals and the Normal or Levy $\alpha$-stable distribution. In agreement with the theory, we can see that the ARMA model fits
well residuals with small amplitude coloured noise, whereas with the increase of $b_{cl}$ the FARIMA model fits better than the ARMA model. Looking at Table 3 in terms of correlation, the Levy $\alpha$-stable distribution fits as good as the Normal distribution as long as the distribution of the residuals is Gaussian without large tails or asymmetry. In Section 2.3, we emphasized that the family of Levy $\alpha$-stable distributions includes the Normal distribution with specific values of its driving parameters (see Equation 4). Thus, the results show that for the amplitude of coloured noise, not very large (i.e. Intermediate - case $B$ - in
Table 2 and 3) compared with the white noise, the two distributions show similar results. However, the scenario with large coloured noise amplitude ($C$), which can generate some aggregation processes thus introducing an asymmetry or large tails





in the distribution of the residuals, emphasizes that the family of Levy $\alpha$-stable distributions perform the best in modelling the residuals' distribution. Note that the asymmetry in the residuals' distribution is relatively limited. Much Larger coloured noise amplitude could produce greater asymmetry in the distribution as seen in financial time series with aggregation processes

of high amplitude (Wooldridge , 2010). Finally, those three scenarios support ideally the theory where in the case of small amplitude coloured noise, the stochastic noise properties are dominated by the Gaussian noise, hence supporting a third r.v. defined as a Gaussian Levy. However, the increase of the coloured noise amplitude shows that it is much more difficult to discriminate between the fractional Levy and the stable Levy. The results point out that the third r.v. can be modelled as a stable Levy process when mostly the FARIMA fits the residuals due to large amplitude long-memory processes, hence creating

a heavy-tail distribution. This result is restrictive for the application to geodetic time series.

### 3.3.2 Real Time Series

We process the daily position time series of three GNSS stations namely $DRAO$, $ASCO$ and $ALBH$ retrieved from the UNAVCO website (UNAVCO , 2009). The functional model includes the tectonic rate, the first and second harmonic of the seasonal signal, and the occurrence time of the offsets. This occurrence time is obtained from the log file of each station.

However, $ALBH$ is known to record slow-slip events from the Cascadia subduction zone (Melbourne et al. , 2005). Thus, we include the offsets provided by the Pacific Northwest Geodetic Array (Miller et al. , 1998). In this scenario we do not know which stochastic model could fit the best the observations. Thus, we use two models: the $PL + WN$ together with the $FN + WN$.

Similar to the previous section, Figure 2 displays the percentage of variations of the stochastic and functional models av-

eraged over the East and North coordinates of each station. Note that the average over the three coordinates is displayed in the appendices (see Figure A1). Because the Up coordinate contains much more noise than the East and North coordinates (Williams et al. , 2004; Montillet et al. , 2013), it amplifies the variation of both stochastic and functional models to several order of magnitude, hence overshadowing the results over the East and North coordinates.

Looking at Figure 2, the first result is that for all the stations, there is a strong dependence with the selected noise model.

When selecting the power-law noise over the flicker noise model, there is an additional variable to estimate (i.e. the power-law noise exponent $\beta$ in Equation (3) ) within the stochastic noise model. Even though our results show a relationship between modelling the residuals and the choice of the stochastic model, our current work does not deal with this issue. Readers interested in this topic can refer to He et al. (2017, 2019).

The second result is the large variations of the functional model compared with the stochastic model. As explained in the

simulations, the functional model partially absorbs the variations of the noise, i.e. the tectonic rate partially fits into the power-law noise. In addition, to some extend at $ASCO$, the sudden increase in the functional model variations at $0.5$ year may be explained due to the absorption of some of the noise with the second harmonic of the seasonal signal.

When comparing the variations of the stochastic and functional models with amplitude below $20\%$ for the stations $DRAO$ and $ASCO$, the results agree with the definition of the fractional Levy process defined in Table 1 as third r.v. modelling the

residuals of the East and North components. The variations of the functional model associated with $ALBH$ are much larger

**Figure 2.** Percentage of variations of the estimated parameters included in the stochastic and functional models when varying the length of the daily position GNSS time series corresponding to the stations $DRAO$, $ASCO$ and $ALBH$. The statistics are estimated over the East and North Coordinates

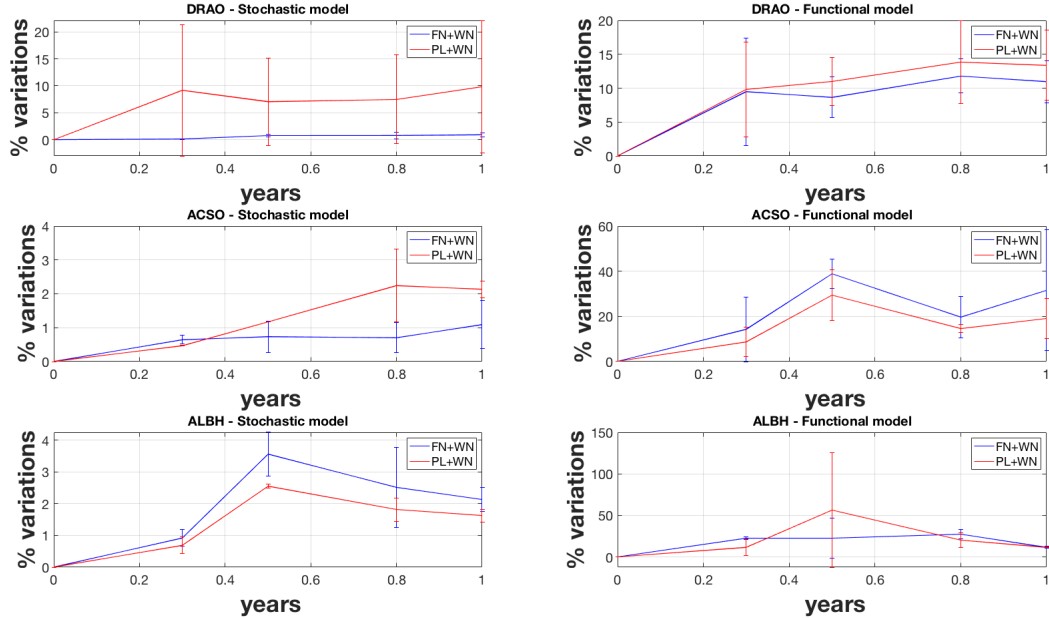

than the other two stations, especially for the $PL+WN$ model with variations up to $50\%$. Those large variations can be explained due to the slow slip events and the difficulty to model the post-seismic relaxations between two consecutive events. In He et al. (2019), the authors justified the selection in the stochastic noise model of a random-walk component together with a $FN+WN$ in order to model the mismatch between the functional model and the observations.

Now Table 4 displays the statistics on the error when fitting the ARMA and FARIMA models to the residuals estimated with the PL+WN stochastic noise model. Note that Table A1 displays in the Appendices the results when using the $FN+WN$ stochastic noise model. The FARIMA and ARMA models perform closely for the whole three stations. The large value for the Up coordinate is due to the amplitude of the noise much larger for this coordinate than for the East and North components (Montillet et al. , 2013). In terms of correlating the distribution of the residuals with the Normal and the Levy $\alpha$-stable dis-

tribution, the correlation value is relatively the same for all stations which indicates that the distribution of the residuals are Gaussian with the absence of large tails. Those results further support the selection of the fractional Levy process as the third r.v. However, the study of real time series also underlines the difficulty to characterize statistically this third r.v.





**Table 4.** Statistics on the Error when fitting the ARMA and FARIMA model to the residual time series for each coordinate of the stations $ALBH$, $DRAO$ and $ASCO$ based on the $PL + WN$ stochastic noise model. Correlation between the distribution of the residuals and the Normal ($Corr.\ Normal$) and the Levy $\alpha$-stable distribution ($Corr.\ Levy$)

| DRAO (PL+WN) | (err. in mm) ARMA | (err. in mm) FARIMA | Corr. Normal | Corr. Levy |
|---|---|---|---|---|
| East | $1.07 \pm 0.01$ | $1.10 \pm 0.07$ | $0.92 \pm 0.05$ | $0.94 \pm 0.05$ |
| North | $1.02 \pm 0.02$ | $1.01 \pm 0.01$ | $0.93 \pm 0.07$ | $0.94 \pm 0.06$ |
| Up | $2.32 \pm 0.21$ | $2.15 \pm 0.30$ | $0.94 \pm 0.04$ | $0.94 \pm 0.05$ |
| ASCO (PL+WN) | | | | |
| East | $0.77 \pm 0.01$ | $0.77 \pm 0.06$ | $0.95 \pm 0.03$ | $0.96 \pm 0.05$ |
| North | $0.84 \pm 0.03$ | $0.73 \pm 0.03$ | $0.97 \pm 0.02$ | $0.96 \pm 0.03$ |
| Up | $2.71 \pm 0.12$ | $2.34 \pm 0.17$ | $0.93 \pm 0.03$ | $0.94 \pm 0.01$ |
| ALBH (PL+WN) | | | | |
| East | $0.97 \pm 0.06$ | $0.87 \pm 0.06$ | $0.94 \pm 0.01$ | $0.94 \pm 0.01$ |
| North | $1.54 \pm 0.03$ | $1.06 \pm 0.14$ | $0.90 \pm 0.02$ | $0.91 \pm 0.04$ |
| Up | $4.36 \pm 0.17$ | $4.08 \pm 0.25$ | $0.92 \pm 0.05$ | $0.94 \pm 0.01$ |

### 3.4 Discussion on the Limits of Modelling with Levy Processes

As discussed in the previous sections, the stable Levy process is characterized by a very large (or infinite) variance. In Montillet
and Yu (2015), it was assumed that the infinite variance of the residual time series comes from large tails of the distribution
(also called heavy tails -(Wooldridge , 2010)), generated by a large amplitude of coloured noise, outliers and other remaining
geophysical signals. The same study implied that the values of the noise variance should be bounded, excluding extreme values.
This is an important assumption to decide whether or not (symmetric) $\alpha$-stable distributions can be used to model any geodetic
time series. Here, we are investigating how the variance due to residual tectonic rate or seasonal signal evolves with the length
of the residual time series (i.e. $L$ epochs).

To recall Section 2.1 and the assumption on the noise properties, let us estimate the mean and variance of the residual time
series. Here, we call the residual time series after the first iteration $\mathbf{s}_1 = [s_1(t_1), ..., s_1(t_L)] = \Delta^1 \mathbf{s}$ as defined in the previous
section. The mean $< s_1(L) >$ and variance $\sigma^2(L)$ are computed over $L$ epochs (i.e. considering the $L$-th epoch defined as $t_L$
$= Ldt$, with the sampling time $dt$ equal 1 for simplification and without taking into account any missing epoch in order to
have a continuous time series). Based on Papoulis and Unnikrishna Pillai (2002), one can estimate the mean over $L$ epochs



$< s_1(L) >$ in the cases of remaining linear trend, such as:

$$
\begin{aligned}
s_1(t_i) &= a_r t_i + b_r + n(t_i) \\
< s_1(L) > &= \frac{1}{L}\sum_{i=1}^{L}(a_r t_i + b_r + n(t_i)) \\
< s_1(L) > &= b_r + a_r \frac{(L+1)}{2} + \mu_C \\
< s_1(L) > &\simeq a_r \frac{L}{2} + \mu_C
\end{aligned}
\tag{6}
$$

where $a_r$ and $b_r$ are the amplitude and the intersect of the residual trend (i.e. remaining tectonic rate). Note that the subscript $r$ designates *residual* in the remaining section. $\simeq$ is the approximation for $L >> 1$. For a time series with $L$ epochs, the variance $\sigma^2(L)$ is:

$$
\begin{aligned}
\sigma^2(L) &= \frac{1}{L}\sum_{i=1}^{L}(s_1(t_i) - < s_1(L) >)^2 \\
\sigma^2(L) &= a_r^2 \frac{(L+1)(2L+1)}{6} - a_r^2 \frac{(L+1)^2}{4} + b_r^2 + \frac{2a_r}{L} Cross + \sigma_n^2(L) - \mu_C(\mu_C + a_r(L+1)) \\
\sigma^2(L) &\simeq \frac{a_r^2 L^2}{12} + \sigma_n^2(L) + b_r^2 - \mu_C a_r L
\end{aligned}
\tag{7}
$$

Note that $Cross$ is the cross term between $a_r t_i$ and the noise term $n(t_i)$. Now, if we assume that the remaining seasonal signal $S_r(t)$ is a pseudo periodic function at frequencies similar to the seasonal signal in Equation (1), hence taking the form $S_r(t) = \sum_{j=1}^{N} c_{r,j}\cos(d_j t) + e_{r,j}\sin(d_j t)$. Thus, we can do the same estimation as above in the case of a remaining pseudo periodic component in the residual time series, such as:

$$
\begin{aligned}
s_1(t_i) &= S_r(t_i) + n(t_i) \\
< s_1(L) > &= \frac{1}{L}\sum_{i=1}^{L}(S_r(t_i) + n(t_i)) \\
< s_1(L) > &\simeq \delta + \mu_C
\end{aligned}
\tag{8}
$$

where $\delta$ is the average of the remaining seasonal signal. It is assumed to be independent of $L$ and bounded such as a periodic function. The variance is equal to:

$$
\begin{aligned}
\sigma^2(L) &= \frac{1}{L}\sum_{i=1}^{L}\sum_{j=1}^{N} c_{r,j}^2 \cos(d_j t)^2 + e_{r,j}^2 \sin(d_j t)^2 + \sigma_n^2(L) \\
&\quad + \frac{2}{L} Cross - < s_1(L) >^2 \\
\sigma^2(L) &\simeq \sigma_n^2(L) + \sum_{j=1}^{N} c_{r,j}^2 + e_{r,j}^2 - (\delta + \mu_C)^2
\end{aligned}
\tag{9}
$$

with $Cross$ is the cross term between $S_r(t)$ and $n(t)$. In the Eq. (6) to (9), the deterministic signals and the noise are assumed completely uncorrelated, which is valid only with white Gaussian noise (i.e. Wiener process) in signal processing (Papoulis





and Unnikrishna Pillai , 2002). As previously discussed in Section 2.1, coloured noise can generate long- memory processes, hence producing non-zero covariance with residual signals. Due to the varying amplitude of the coloured noise in geodetic time series with mixed spectra, the uncorrelated assumption is currently debated within the community (Herring et al. , 2016; He et al. , 2017). Therefore, recent works have introduced a random component together with a deterministic signal: nonlinear rate (Wang et al. , 2016; Dmitrieva et al. , 2017), non-deterministic seasonal signal (Davis et al. , 2012; Chen et al. , 2015; Klos et al. , 2018). Thus, strictly speaking, $\sigma^2$ should be seen as an upper bound.

The closed-form solution of the variance $\sigma^2(L)$ shows that the variance is unbounded in the case of a residual linear trend. To recall the discussion in Section 3.1, if this residual trend originates from various sources not well-described in the functional and stochastic model (i.e. undetected jumps, small amplitude random-walk component) of the geodetic time series, the amplitude of this trend should be rather small ($a < 1$ mm/yr) considering the length of GNSS time series available until now ($L < 30$ years). Unless this nonlinear residual trend has a large amplitude, a correction of the functional model must be done a posteriori due to possible anxiety between the models and the observations. The same remarks can be applied to the variance of the remaining seasonal signal where a large amplitude would imply a misfit with the functional model. Thus, we expect rather small amplitude of the coefficients $c_{r,j}$ and $e_{r,j} \sim 0.1$ to $\sim 0.001$ mm. Also, in the Appendix $B$, we have developed a similar formula to take into account undetected offsets, where we show that the variance is also bounded. In this case, a large value would mean that one or several large offsets have not been included in the functional model.

## 4 Conclusions

We have investigated the statistical assumptions behind using the fBm and the family of $\alpha$-stable distributions in order to model the stochastic processes within the residual GNSS time series. We model the residual time series as a sum of three random variables (r.v.). The first two are defined from the stochastic model and assumptions on the noise properties of the geodetic time series. The third r.v. is assumed to belong to the Levy processes. We then distinguish three cases. In the case of a residual time series containing only short-term processes, the r.v. is a Gaussian Levy process. In the presence of long-term correlations and exhibiting self-similarity property, fractional Levy processes can be seen as an alternative model of using the fBm. Due to the linear relationship between the Hurst parameter and the fractional parameter of the FARIMA, it is likely that the FARIMA can fit the residual time series under specific conditions (i.e. amplitude of the coloured noise). The third case is the stable Levy process, with the presence of long-term correlation processes, high amplitude aggregation processes or random-walk.

In order to check our model, we have simulated mixed spectra time series with various levels of coloured noise. We have then developed a $N$ steps methodology based on varying the length of the time series (limited to 1 year) to study the variations of the stochastic and functional models and to model the distribution of the residuals. The results emphasize the difficulty to separate the fractional Levy process and the stable Levy process mainly due to the absorption of the variations of stochastic processes by the functional model, unless the distribution of the residuals exhibits heavy-tails. Another difficulty is the dependence of the results with the stochastic noise model. The use of real GNSS time series supports the results based on simulated ones.

However, the discussion on the limits of modelling the stochastic properties of the residuals with the stable Levy process



underlines that the infinite variance property can only be satisfied in the case of heavy-tailed distributions. This condition is
generally satisfied if there is a large amplitude random-walk (e.g., temporal aggregation in financial time series) or an important
misfit between the models (i.e. functional and stochastic) and the observations, which means that there is anxiety in the choice
of the functional model (e.g., unmodelled large jumps, large outliers). Finally, with longer and longer time series, one may be
able to statistically characterize more precisely the third r.v.

*Acknowledgements.* We would like to thank Dr. Machiel S. Bos from the SEGAL for multiple discussions on the stochastic properties of
the GNSS time series. Dr. Xiaoxing He was sponsored by the Doctoral Fund of Ministry of Education of China (2018M632909) and the
National Natural Science Foundation of China (41574031).

Appendix A

**Appendix:  fBm and fLsm: integral representation**

The fractional Brownian motion (fBm) $\{B_H(t)\}_{t \geq 0}$ has the integral representation:

$$B_H(t) = \int_{-\infty}^{\infty} \left( (t-u)_+^{H-\frac{1}{2}} - (-u)_+^{H-\frac{1}{2}} \right) dB(u) \tag{1}$$

where $x_+ = max(x, 0)$ and $B(u)$ is a Brownian motion (Bm). It is $H$-self-similar with stationary increments and it is the only
Gaussian process with such properties for $0 < H < 1$ (Samorodnitsky and Taqqu , 1994).

From Weron et al. (2005), the fractional Levy stable motion (fLsm) can be defined with the process $\{Z_\alpha^H(t)\}$ (with $t$ in $\mathbb{R}$)
by the following integral representation:

$$Z_\alpha^H = \int_{-\infty}^{\infty} \left( (t-u)_+^{H-\frac{1}{\alpha}} - (-u)_+^{H-\frac{1}{\alpha}} \right) dZ_\alpha(u) \tag{2}$$

where $Z_\alpha(u)$ is a symmetric Levy $\alpha$ -stable motion (Lsm). The integral is well defined for $0 < H < 1$ and $0 < \alpha \leq 2$ as a
weighted average of the Levy stable motion $Z_\alpha(u)$. The process $\{Z_\alpha^H(t)\}$ is H-self-similar and has stationary increments.
Comparing the definition of fBm and fLsm, we can observe that fLsm is similar to fBm for the case $\alpha = 2$.

Appendix B





**Appendix: Estimation of the Variance in the Presence of Offsets**

We model here the offsets in the time series as Heaviside step functions according to He et al. (2017). Following Section 3.4, the residual time series in presence of remaining offsets can be written such as

$$s_1(t_i) = \sum_{k=1}^{ng} g_k \mathcal{H}(t_i - T_k) + n(t_i) \tag{1}$$

Where $\mathcal{H}$ is the Heaviside step function. One can estimate the mean over $L$ epochs:

$$
\begin{aligned}
<s_1(L)> &= \frac{1}{L}\sum_{i=1}^{L}(\sum_{k=1}^{ng} g_k \mathcal{H}(t_i - T_k)) + \mu_C \\
<s_1(L)> &= \frac{1}{L}\sum_{k=1}^{ng} g_k \mathcal{H}(t_L - T_k) + \mu_C
\end{aligned}
\tag{2}
$$

The variance is equal to

$$
\begin{aligned}
\sigma^2(L) &= \frac{1}{L}\sum_{i=1}^{L}(\sum_{k=1}^{ng} g_k \mathcal{H}(t_i - T_k) + n(t_i) - <s_1(L)>)^2 \\
\sigma^2(L) &\simeq \sigma_n^2(L) + \frac{1}{L}(\sum_{k=1}^{ng} g_k \mathcal{H}(t_L - T_k))^2 - <s_1(L)>^2
\end{aligned}
\tag{3}
$$

In the presence of small (undetectable) offsets ( $g_k < 1$ mm), we can further assume that $<s_1(L)> \sim \mu_C$ and $\sigma^2(L) \sim \sigma_n^2(L) - \mu_C^2$. For multiple large uncorrected offsets (i.e. noticeable above the noise floor), the variance can be large, but the distribution of the residual time series should look like various Gaussian distributions overlapping each other corresponding to the segments of the time series defined by those noticeable offsets. This case is not taken into account in our assumptions summarized in Table 1, because it supposes that there is a large anxiety about the chosen functional model - obviously missing some large 425 offsets.





Appendix C

## Appendix: Additional Results

**Figure A1.** Percentage of variations of the estimated parameters included in the stochastic and functional models when varying the length of the daily position GNSS time series corresponding to the stations $DRAO$, $ASCO$ and $ALBH$. The statistics are estimated over the East, North and Up Coordinates

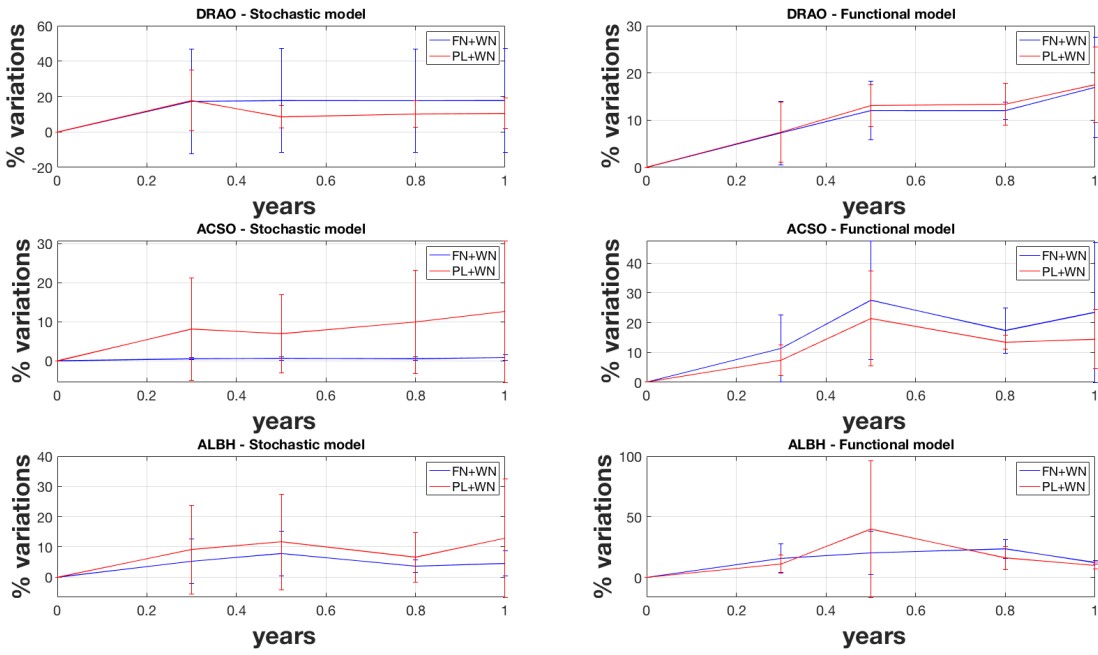





**Table A1.** Statistics on the Error when fitting the ARMA and FARIMA model to the residual time series for each coordinate of the stations $ALBH$, $DRAO$ and $ASCO$ based on the $FN + WN$ stochastic noise model. Correlation between the distribution of the residuals and the Normal ($Corr.\ Normal$) and the Levy $\alpha$-stable distribution ($Corr.\ Levy$)

| $DRAO\ (FN + WN)$ | (err. in mm) ARMA | (err. in mm) FARIMA | Corr. Normal | Corr. Levy |
|---|---|---|---|---|
| East | $1.07 \pm 0.01$ | $1.00 \pm 0.02$ | $0.92 \pm 0.03$ | $0.94 \pm 0.05$ |
| North | $1.02 \pm 0.02$ | $1.32 \pm 0.07$ | $0.92 \pm 0.05$ | $0.94 \pm 0.04$ |
| Up | $2.33 \pm 0.18$ | $2.20 \pm 0.32$ | $0.94 \pm 0.08$ | $0.94 \pm 0.05$ |
| $ASCO\ (FN + WN)$ | | | | |
| East | $0.77 \pm 0.01$ | $0.75 \pm 0.07$ | $0.95 \pm 0.02$ | $0.96 \pm 0.01$ |
| North | $0.85 \pm 0.03$ | $0.74 \pm 0.05$ | $0.94 \pm 0.01$ | $0.96 \pm 0.01$ |
| Up | $2.18 \pm 0.14$ | $2.51 \pm 0.21$ | $0.93 \pm 0.03$ | $0.94 \pm 0.03$ |
| $ALBH\ (FN + WN)$ | | | | |
| East | $0.97 \pm 0.04$ | $0.86 \pm 0.06$ | $0.93 \pm 0.01$ | $0.94 \pm 0.01$ |
| North | $1.52 \pm 0.08$ | $1.08 \pm 0.10$ | $0.91 \pm 0.02$ | $0.91 \pm 0.04$ |
| Up | $3.83 \pm 0.21$ | $3.32 \pm 0.15$ | $0.93 \pm 0.03$ | $0.94 \pm 0.01$ |

## Appendix: References

Benassi, A., Cohen, S., and Istas, J.: On roughness indices for fractional fields, Bernoulli, 10(2), 357-373, 2004.

Bock, Y., Melgar, D.: Physical applications of GPS geodesy: a review. Rep. Prog. Phys., 79 (10)., doi:10.1088/0034-4885/79/10/106801, 2016.

Bos, M.S., Fernandes, R.M.S., Williams, S.D.P., Bastos, L.: Fast error analysis of continuous GNSS observations with missing data, J. Geod., 87, 351–360. doi.org/10.1007/s00190-012-0605-0, 2013.

Botai, O.J., Combrinck, L., Sivakumar, V.: Interferences of $\alpha$-stable distribution of the underlying noise components in geodetic data, South
Afr. J. of Geo.. doi:10.2113/gssajg.114.3-4.541, 2011.

Chen, Q., Weigelt, M., Sneeuw, N., van Dam, T.: On Time-Variable Seasonal Signals: Comparison of SSA and Kalman Filtering Based Approach, in: Sneeuw N., Novak P., Crespi M., Sanso F. (eds) VIII Hotine-Marussi Symposium on Mathematical Geodesy. International Association of Geodesy Symposia, vol 142. Springer, Cham, 2015.

Cont, R., Tankov, P.: Financial modelling with jump processes, Chapman & Hall/CRC, ISBN 1-58488-413-4, 2004.

Davis, J.L., Wernicke, B.P., Tamisiea, M.E.: On seasonal signals in geodetic time series. J. Geophys. Res., 117 (B01403). doi:10.1029/2011JB008690, 2012.

Dmitrieva, K., Segall, P., Bradley, A. M.: Effects of linear trends on estimation of noise in GNSS position time-series, Geophys. J. Int., 208(1), 281-288, doi:10.1093/gji/ggw391, 2017.

Granger, C.W., Joyeux, R. : An introduction to long-memory time series models and fractional differencing, J. Time Ser. Anal., 1, 15-29,
445 1980.





Gazeaux, J., Williams, S., King, M., Bos, M., Dach, R., Deo, M., Moore, A. W., Ostini L., Petrie, E., Roggero, M., Teferle, F. N., Olivares, G., Webb, F. H.: Detecting offsets in GPS time series: First results from the detection of offsets in GPS experiment, J. Geophys. Res., 118(5), 2397 - 2407, 2013.

Haykin, S., Adaptive Filter Theory. fourth edition, Prentice Hall Upper Saddle River, New Jersey, 2002.

He, X., Montillet, J.-P, Fernandes, R.M.S., Bos, M.S., Yu, K., Hua, X., Jiang, W.: Review of current GPS methodologies for producing accurate time series and their error sources, J. Geodyn., 106. doi:10.1016/j.jog.2017.01.004, 2017.

He, X.,Bos, M.S., Montillet, J.-P., Fernandes, R.M.S.: Investigation of information criteria and noise models for GNSS time series, J. Geod., doi: 10.1007/s00190-019-01244-y, 2018.

Herring, T.A., King, R.W., McClusky, S.C.: Introduction to GAMIT/GLOBK, report, MIT, Cambridge, 2010.

Herring, T.A., King, R.W., McClusky, S.C., Floyd, M., Wang, L., Murray, M., Melbourne, T.,Santillan, M., Szeliga, W., Phillips, D., Puskas, C.: Plate Boundary Observatory and Related Networks: GPS Data Analysis Methods and Geodetic Products, Rev. Geophys.,54. doi:10.1002/2016RG000529, 2016.

Kasdin, N.: Discrete simulation of colored noise and stochastic processes and $1/f^{\alpha}$ power-law noise generation, Proc. IEEE, vol. 83, 1995.

Klos, A., Bos, M.S., Bogusz, J.: Detecting time-varying seasonal signal in GPS position time series with different noise levels, J. GPS Solut.,

22 (21), doi:10.1007/s10291-017-0686-6, 2018.

Koutrouvelis, I.A.: Regression-type estimation of the parameters of stable laws, J. Am. Statist. Assoc., 75:918-928, 1980.

Langbein, J.: Noise in GPS displacement measurements from Southern California and Southern Nevada, J. Geophys. Res., 113, B05405, doi:10.1029/2007JB005247, 2008.

Langbein, J., Svarc, J. L.: Evaluation of Temporally Correlated Noise in Global Navigation Satellite System Time Series: Geodetic Monument

Performance, J. Geophys. Res., 124(1), 925-942, 2019.

Li, Q., Tao, B.: Theory of Probability and Statistics, and Its Applications to Geodesy, Survey and Mapping, 321, 1982.

Li, J., Miyashita, K., Kato, T., Miyazaki, S.: GPS time series modeling by autoregressive moving average method: Application to the crustal deformation in central Japan, Earth Planet Space, 52, 155-162, 2000.

Li, T.-H.: Time Series with Mixed Spectra, CRC Press, ISBN 9781584881766, 2013.

Lilly, J.M., Sykulski, A.M., Early, J. J., and Olhede, S. C.: Fractional Brownian motion, the Matérn process, and stochastic modeling of turbulent dispersion, Nonlin. Processes Geophys. 24, 481–514. doi:10.5194/npg-24-481-2017, 2017.

Mandelbrodt, B., Van Ness, J.W.: Fractional Brownian Motions, Fractional Noises and Applications, SIAM Rev., 10(4):422-437, 1968.

Melbourne, T. I., Szeliga, W. M., Miller, M., Santillan, V. M.: Extent and duration of the 2003 Cascadia slow earthquake, Geophys. Res. Lett., 32, L04301. doi:10.1029/2004GL021790, 2005.

Miller, M. M., Dragert, H., Endo, E., Freymueller, J. T., Goldfinger, C., Kelsey, H. M., et al.: PANGA: Precise measurements help gauge Pacific Northwest's Earthquake Potential, Eos Transactions, American Geophysical Union, 79(23), 269–275, 1998.

Montillet, J.-P., Tregoning, P., McClusky, S., Yu, K.: Extracting white noise statistics in GPS coordinate time series, IEEE Geosci. Remote Sens. Lett., 10(3):563-567, doi:10.1109/LGRS.2012.2213576, 2013.

Montillet, J.-P., Williams, S.D.P., Koulali, A., McClusky, S.C.: Estimation of offsets in GPS time-series and application to the detection of

earthquake deformation in the far-field, Geophys. J. Int., 200(2), 1207-1221. doi.org/10.1093/gji/ggu473, 2015.

Montillet, J.P., Yu, K.: Modeling geodetic processes with levy $\alpha$-stable distribution and FARIMA. Math. Geosci., 47(6). doi:10.1007/s11004-014-9574-6, 2015.



Montillet, J.-P., and Bos, M.S.: Geodetic Time Series Analysis in Earth Sciences, Springer Geophysics, doi: 0.1007/978-3-030-21718-1, 2019.

Nikias, C.L., Shao, M.: Signal processing with Alpha-Stable Distributions and Applications, New York, Wiley edition, 1995.

Nolan, J.P.: Stable Distributions - Models for heavy tailed data. Book online: $http://www.math.ucla.edu/biskup/275b.1.13w/PDFs/Nolan.pdf$, 2009.

Panahi, H.: Model Selection Test for the Heavy-Tailed Distributions under Censored Samples with Application in Financial Data, Int. J. of Financial Studiesb(MDPI), 4(4), doi:10.3390/ijfs4040024, 2016.

Panas, E.: Estimating fractal dimension using stable distributions and exploring long memory through ARFIMA models in Athens Stock Exchange, Appl. Fin. Econ., 11(4):395-402, 2001.

Papoulis A., Unnikrishna Pillai S. (2002) Probability, Random Variables and Stochastic Processes, 4th Ed., McGraw-Hill Series in Electrical and Computer Engineering, the McGraw-Hill companies. ISBN: 0-07-366011-6, 2002.

Samorodnitsky, G., and Taqqu, M. S.: Stable Non- Gaussian Random Processes: Stochastic Models with In- finite Variance (Chapman and Hall, London), 1994.

Sowell, F.: Modeling long-run behavior with the fractional ARIMA model, J. Monetary Econ., 29, p.277-302, 1991.

Tregoning, P., Watson, C.: Atmospheric effects and spurious signals in GPS analyses. J. Geophys. Res., 114 (B09403). doi:10.1029/2009JB006344, 2009.

UNAVCO: Plate Boundary Observatory: The first five years. Boulder, CO: UNAVCO. Retrieved from $https://www.unavco.org/education/outreach/pamphlets/2009-PBO/PBO-2009-brochure-first-five-years.pdf$, 2009.

Wang, C., Liao, M., Li X.: Ship Detection in SAR Image Based on the Alpha-Stable Distribution, Sensors, 8:4948-4960. doi:10.3390/s8084948, 2008.

Wang, X., Cheng, Y., Wu, S., Zhang, K.: An enhanced singular spectrum analysis method for constructing non-linear model of GPS site movement, J. of Geophys. Res., 121 (3), doi:10.1002/2015JB012573, 2016.

Weron, A., Burnecki, K., Mercik, S., and Weron, K.: Complete description of all self-similar models driven by Levy stable noise, Phys. Rev. E., doi: 10.1103/PhysRevE.71.016113, 2005.

Williams, S.D.P.: The effect of coloured noise on the uncertainties of rates estimated from geodetic time series, J. Geod., 76, p.483-494, 2003.

Williams, S.D.P., Bock, Y., Fang, P., Jamason, P., Nikolaidis, R.M., Prawirodirdjo, L., Miller, M., Johnson, D.J.: Error analysis of continuous GPS position time series, J. Geophys. Res., 109(B03412), doi:10.1029/2003JB002741, 2004.

Wooldridge, J.M.: Econometric Analysis of Cross Section and Panel Data. First edition, MIT Press, ISBN (13):9780262232586, 2010.

Working, H.: Note on the Correlation of First Differences of Averages in a Random Chain. Econometrica, 28(4), 916-918. doi:10.2307/1907574, 1960.