# Peer review of "Application of Levy Processes in Modelling (Geodetic) Time Series With Mixed Spectra"

_Nonlinear Processes in Geophysics, 2019_

## Referee Comment (RC1) · Anonymous Referee #1 · 2 Dec 2019

The paper discusses using Gaussian and non-Gaussian processes for geodetic time-series. The topic is very nice and timely, as there is a wide interest in community for this kind of processes, and hence all contributions are welcome. The current paper has a literature review on the models, and builds up a case study on specific synthetic and real datasets.

The drawback of the paper is that it does not provide a solid mathematical framework on which the time-series analysis is carried out. In my experience, this kind of papers should always have four key elements: Statistical model, estimation algorithm, synthetic examples and real data examples. Unfortunately the statistical model is not

properly build, and several mathematical errors are in the text. These could be alleviated by writing out formulas explicitly, so that the authors and readers could understand the model. The major issues of the paper are:

1) There are repeated fundamental mathematical mistakes and misunderstandings in the paper, including claim that wide sense stationary can have a temporally evolving mean, and that Brownian motion is stationary process (or that's at least how the reader understands the text). As the Lévy processes are mathematically extremely technical, the simple Gaussian process definitions cannot have such fundamental flaws. For example, random variable is used in many places where stochastic process should be used – Stochastic processes are collections of random variables.

2) For the parameter estimation algorithm, a 'Hector software' is used. We need to have more details than just the name of software and citation. We need to know whether this is optimisation or stochastic based estimation algorithm, and what kind of output it produces.

3) For the synthetic and real time-series analysis the authors do not show the typical time-series they are dealing with, but just some kind of summary statistics, which is totally insufficient. This makes it impossible to understand the benefit of the constructed method.

As the paper more or less uses widely known models, the paper is at best incremental from methodology point of view. The language used has lots of grammar problems. This is highlighted by the fact that authors use Levy instead of Lévy already in the paper title. There are lots of editorial problems, like exhaustive bracket using. I list below more several smaller and larger issues. Because of the underdeveloped results and presentation, I must, unfortunately, recommend rejection.

Some comments on the text:

Abstract: "Stable process is characterized by a large variance" — alpha-stable processes do not have variances, rephrase.

line 14: bracket before e.g., but no closing bracket in the sentence.

line 16: extra hyphen before references?

line 17: What do you mean by functional models? Possibly parametric models?

line 25: flicker noise has power spectrum $1/f$, and thus it does not have a covariance function, and hence should be treated as a generalised stochastic process – simple mentioning of non-stationarity is not sufficient here.

line 27: Mandelbrodt –> Mandelbrot (also elsewhere)

line 58: cos and sin should not be in italic.

Equation (1) — $c_j$ and $e_j$ are not defined

Equation (2) — Use regular transpose notation, e.g. $T$ and not dagger. Notation is sloppy, is $n0$ actually $n_0(t)$? ... and similarly $n1$!?

Line 67: What do you mean by "variance-covariance matrix" – define everything.

Paragraph starting line 71: If you make the assumption of wide-sense stationarity, you cannot assume a non-constant mean. Please check your definitions, and adjust.

Section 2.2: This section needs to rewritten by using mathematical formulas showing the relations between different objects – it is impossible now to follow this text.

Equation (4) 'sign' should not be in math format, there should be space after the second if, and ifs should be in text mode.

Line 122: fBm is self-similar, not all self-similar are fBm. Please be careful on the wording. Paragraph starting line 128: Please put the equations to the text – it is not enough to have some elementary formulas in the appendix. Also, do note that you cannot extend from fBm to fLsm by simple use of $H = 1/\alpha$, as you need to also change the probability measures. Please discuss this in detail, and discuss what is

actually needed mathematically to have a properly defined fLsm.

Paragraph starting line 133: Please do write out the equations. This is too sloppy.

Line 143: You are dealing with stochastic processes, not random variables, adjust the language.

Line 144: a N-steps method —> an N-step method

Line 148: Space missing between model(Langbein

Enumerated list in page 6: Please use 1., 2. and 3. instead of 1-, 2- and 3-. Also item 1- has wrong properties, e.g. Brownian motion is not stationary. Also please check terminology in other parts, e.g. in 2- you talk about Gaussian distribution (one-dimensional) and then relate that to a stochastic process. Please see all these through.

Line 166: presents –> present

Section 3.2: Here in the beginning, you abandon all the non-Gaussian models discussed earlier, and return to a Gaussian case. Why?

Line 186: You should include the whole model in your model description, and not leave some parts in appendix – this includes also offset modelling, which is extremely important to include in the statistical analysis, as otherwise the methodology is not transparent.

Equation (5): Ok, here I am totally lost — you have an additive model, and you have somehow got the parameters $\widehat{\theta}_1$ and $\widehat{\theta}_2$, but how do you get them? What is your estimation algorithm, MCMC, optimisation, something else? Then you compute an extrapolation $[s_{L+1},...,s_{L+N}]$. Why? What is N-th variation over here. You have two definitions for s here, and then you have a number of objects which you do not define, and use $\Delta^N$ sign ... is this N-th derivative?

Line 218: "Moreover, the estimation of the model parameters is carried out using the Hector software" — please open up this. Is it MCMC, optimisation or what kind of

toolbox. I have never heard about it, so some basic background should be needed.

Line 234: defacto –> de facto

Section 3.3.1 – It is impossible to assess this one, as the time-series are not shown. Please plot all the necessary standard things for full understanding of the model. The current level is not adequate.

Section 3.3.2 — The same comments as for Section 3.3.1

Line 319: Very large variance for Lévy process?

Line 328: Please use $\langle \cdot \rangle$ instead of $< \cdot >$

Line 337: Please use $\gg$ not $>>$

line 356: long- memory –> long-memory

---

## Author Comment (AC1) · 6 Dec 2019

Dear Reviewer,
Thank you for taking time to comment the manuscript. A few comments may be a general criticism due to the "jargon" used specifically in geodetic time series which sometimes is not as precise as in pure statistics. Thus, we would like to discuss deeply the major comments.

[Figure]

**1 General comment "The drawback of the paper is that it does not provide a solid mathematical framework on which the time-series analysis is carried out."**

We would like to emphasise that this manuscript focuses on modelling the stochastic noise processes of the geodetic time series using 3 stochastic processes, instead of 2 as commonly assumed. We justify our approach considering that recent papers (i.e. He et al., 2019; Book Montillet & Bos 2019; Langbein & Vrac 2019) have introduced the model Flicker Noise + White Noise + Random Walk ($FN + WN + RW$) to model these stochastic processes. Note that we will correct the manuscript by rephrasing the misuse of "random variable" instead of "stochastic process" as suggested by the reviewer.

**2 "Unfortunately the statistical model is not properly build ..."**

The approach taken in this paper follows the fitting of a trajectory/functional model together with a stochastic noise model. That is a comparable approach as the "parametric" approach defined in applied statistics. In Montillet et Bos (2019) (chapter 1), this topic is comprehensively discussed. Unfortunately, due to a lack of space, we have only reminded in the introduction to the reader the basic elements of this approach. Of course, there are different alternatives to this approach. One of them is the general "non-parametric" approach (e.g., MIDAS, $doi.org/10.1002/2015JB012552$). In the final manuscript, corrections will be made to the manuscript in order to avoid any misunderstanding.

Now, in order to model the stochastic noise properties of the geodetic time series, we study the stochastic noise properties of the residual time series, that is the time series after subtracting the estimated functional/trajectory model (e.g., tectonic rate and sea-

sonal signal). That is why we have developed this N-step algorithm which is basically the iterative estimation of the functional and stochastic noise models when increasing the time series length. This algorithm allows investigating the variations of the estimated stochastic parameters, and thus to conclude on the proposed assumptions of a third stochastic process.

Therefore, we have selected three cases for the 3rd stochastic process: Gaussian, Fractional Levy and Stable Levy.

1- The Gaussian case is obvious, which is the case that the only important changes in the stochastic noise properties are in the parameters related to the white noise or slight variations of the coloured noise amplitude.

2-The Fractional Levy is related to the fractional Levy stable motion (fLsm) which is related to the fBm based on Weron et al. 2005. The relationship with the fBm is important, because several works on modelling geodetic time series are using the fBm, which then justifies this assumption. That is generally the variations of the coloured noise properties. If so, it is generally due to unmodeled transient signals, or small offsets buried in the noise floor. A particular case is the use of the Random-walk in the stochastic noise model as previously discussed.

The last case is the alpha stable process which is related to model the residual time series with the family of alpha-stable Levy distributions. That is a very specific case, because we underline in the conclusions that this case happens when residual time series are modelled by heavy tailed distributions.

**3 "As the Lévy processes are mathematically extremely technical, the simple Gaussian process definitions cannot have such fundamental flaws."**

In the light of the above explanations, we have restricted the use of the Levy processes to these 3 cases. Therefore, a full discussion on the Levy processes is out of the scope of this paper. In the supplementary materials, we will add a specific section on general definition of Levy processes, but it should remain outside of the main body of the manuscript to avoid any confusions.

**4 "For the parameter estimation algorithm, a 'Hector software' is used" (This comment also refers to the use as Hector without its statistical basis)**

The Hector software is based on maximum-likelihood such as:

$$\ln(L) = -1/2[N \ln(2\pi) + \ln(det(\mathbf{C})) + (\mathbf{y} - \mathbf{Ax})^T \mathbf{C}^{-1}(\mathbf{y} - \mathbf{Ax})] \tag{1}$$

With $(\mathbf{y} - \mathbf{Ax})$ represent the observations minus the fitted model and are normally called the residuals. The function $ln(L)$ must be maximised assuming that the covariance matrix $\mathbf{C}$ is known. There has been an important effort produced by geodesists to define $\mathbf{C}$ which is expressed by a long literature from the early works (Williams et al., 2004) to current discussions (see Montillet and Bos, 2019, Chapter 2 for a full summary). Now, the current definition of the covariance matrix $\mathbf{C}$ related to the sum of white and coloured noise is as expressed in equation 2

$$\mathbf{C} = \sigma_{n_0}^2 \mathbf{I} + \sigma_{n_1}^2 \mathbf{J} \tag{2}$$

Where I is the identity matrix, and J is the approximation of the covariance matrix of the selected coloured noise. This approximation depends on the software used to do

the maximisation of the function $\ln(L)$. Generally, the main assumption is to use an approximation of power-law noises. Granger and Joyeux (1980) and following studies (see Montillet and Bos, 2019 Chapter 2 for more information) demonstrated that power-law noise can be achieved using fractional differencing of Gaussian noise. Several approximations can then be derived in order to simplify to perform the estimation of the inverse of $\mathbf{C}$, which quickly becomes a large matrix for long geodetic time series ($> 5$ years $= 1825$ samples), and taking into account the large amount of data when dealing with large network of stations ($> 100$ stations). In the specific case of Hector, Bos et al. (2013) made several assumptions on the coloured noise properties in order to get a Toeplix matrix to define J. Note that in the case of a third stochastic process, a third covariance matrix is added to equation 2. In order to improve the manuscript, we will add this paragraph to section 2.1.

**5 " For the synthetic and real time-series analysis the authors do not show the typical time-series they are dealing with, but just some kind of summary statistics, which is totally insufficient."**

To recall the abstract of the paper, the interest of our development is supporting the approach of the three stochastic processes with different cases. Thus, we will add simulated time series in the appendices to give more details and support the simulations. We must emphasise that the Hector software includes a package to simulate geodetic time series (see Hector manual – see example 3 - $http : //segal.ubi.pt/hector/manual_1.7.2.pdf$).

**6  "As the paper more or less uses widely known models, the paper is at best incremental from methodology point of view"**

One must take into account that this discussion is part of a hot topic in geodetic time series analysis and a general effort to model more precisely longer and longer time series. For example, in He et al. (2019), the authors found for some time series that very long geodetic time series (over 10 years), the PSD experienced a flattening in the high frequency which makes the power-law model not the best choice anymore to model long-term stochastic processes. Thus, we believe that this research fits well into this general debate in geodesy, and the use of these Levy processes can shed a new light on the fundamental approach of modelling stochastic noise processes in geodetic time series.

**7  "Section 3.2: Here in the beginning, you abandon all the non-Gaussian models discussed earlier, and return to a Gaussian case. Why?"**

There is a confusion here. We do explain first the N-Step algorithm (see clarifications below). Then, we show how this algorithm allows to discriminate between the three cases for the third stochastic process. The Gaussian and non-Gaussian cases are explained together with Table 1.

**8  "If you make the assumption of wide-sense stationarity, you cannot assume a non-constant mean"**

A random process is said to be following the WSS assumption if its mean and auto-correlation function are time invariant (Haykins, 2000). Based on this definition, we

cannot strictly introduce a slow varying mean. Therefore, we need to revise section 2.1 accordingly.

**9 " stable Levy process with very large (infinite) variance?"**

Here, we assume that the stable Levy process is an alpha-stable Levy motion, selected as the third stochastic processes if the distribution of the residual time series displays heavy tails, therefore with a very large variance. We do not imply that every stable Levy process is characterised by a large variance. We will clarify this sentence in the manuscript.

**10 Clarification on the algorithm – the N-step process**

"Equation (5): Ok, here I am totally lost — you have an additive model, and you have somehow got the parameters $b_1$ and $b_2$, but how do you get them? What is your estimation algorithm, MCMC, optimisation, something else? Then you compute an extrapolation $[sL + 1, \ldots, sL + N]$. Why? What is N-th variation over here. You have two definitions for s here, and then you have a number of objects which you do not define, and use $_N$ sign ... is this $N$-th derivative?"

The algorithm is an iteration of the estimation of the functional + stochastic noise model using Hector. The iteration is defined by taking longer and longer time series at each step, or in other words adding more data sample at each iteration. As mentioned earlier on, Hector is a software based on maximum likelihood. The first step estimates the time series with $L$ data samples, whereas at the $N$-th iterations there are $L + N$ samples. The general idea is to see the variations in the estimated parameters of the stochastic

noise model in order to select which type of processes (one of the 3 Levy processes) best models the stochastic noise properties.

In details, the algorithm is defined in a functional form. Thus, the time series is a sum of :

- $F(\theta_1)$ : the trajectory model, which contains the functions describing various geo-physical signals (seasonal signal, tectonic rate, offsets, post-seismic relaxations ...) - discussed in equation 1

- $G(\theta_2)$: the stochastic noise model (Flicker + white noise, Power-law + white noise ...)

$\theta_1$ contains for example the amplitude of the seasonal signal, the amplitude of the tectonic rate, the time and amplitude of various offsets (i.e. related to geophysical phenomena or equipment changes), the estimation of the amplitude and time of a post-seismic relaxation. $\theta_2$ contains the amplitude of the white noise, the amplitude of the coloured noise (depending on the selected model), the power-law index (if it is selected as an unknown variable). The parameters also vary with the stochastic model selected.

At the N-th iteration, we get the residual time series $\Delta S^N$ by subtracting the time series $S$ with the estimated functional model $[F(\hat{\theta_1})]_N$. We then look at the estimated parameters ($\hat{\theta_2}$ ) in the stochastic noise model $[G(\hat{\theta_2})]_N$ and compared their variations with the previous estimates in previous iterations. Figure 1 and 2 show these variations using simulated and real time series. The $_N$ sign refers to the $N$-th iteration - the time series with $L + N$ samples.

Note there are no objects (no derivatives), just parameters described in Section 2. Also, we do not "extrapolate" the time series. Instead we use the time series by varying the length on the last year. We choose only one year (and no more), because we state

at Line 191 :

"(. . .) varying the length of the time series over several years is not realistic taking into account that real time series can record undetectable transient signals, undocumented offsets and other non-deterministic signals unlikely to be modelled precisely " The final version of the manuscript should include a clarification of the algorithm in the light of this discussion.

**11   "Section 2.2: This section needs to rewritten by using mathematical formulas showing the relations between different objects – it is impossible now to follow this text."**

We will revise this section by adding a proper definition of the ARMA and FARIMA. Note that the general idea is that the fitting of the FARIMA to the residual time series implies long-memory correlation.  With the relationship with "d", parameter of the FARIMA and "H" (the Hurst parameter), there is a direct relationship between the FARIMA and the fBm (Montillet and Yu, 2015). Now, taking into account that fLsm is also defined with "H", we have also a direct relationship between FARIMA and fLsm. Also, we should clarify the relationship between the fBm and the fLsm as proposed by the reviewer.

Please also note the supplement to this comment:
https://www.nonlin-processes-geophys-discuss.net/npg-2019-48/npg-2019-48-AC1-supplement.pdf

---

## Referee Comment (RC2) · Anonymous Referee #2 · 18 Mar 2020

This manuscript discusses the use of different scaling processes to model geodetic time series. The logical structure of the manuscript is so confuse that I can only recommend to reject the manuscript. Here are comments and suggestions.

- First of all the authors seem to have published similar works recently (He et al., 2017, 2018; Montillet and Yu, 2015): the authors should indicate clearly what is new in this manuscript with respect to these previous works - the authors do not introduced adequately the topic: what is GNSS data, why does it have non-stationary and stochastic components? - in section 2.2 the authors indicate that GNSS time series have a scaling power spectrum with slope beta. But no data is shown to justify this. All the structure

of the manuscript is very confusing, with many different models proposed from time to time, with no justification from the data. It really seems a mess. - a more orderly and correct structure would be the following: first explain what is GNSS data and why it is non stationary and stochastic. Then show some plots of such data, with power spectrum and pdf. Then this justifies the use of given modelling with given hypotheses and parameters. Then, explain how to estimate the parameters.

Other comments: - the citations to previous works is too erratic, with bad placed spaces before commas. The authors should be much more cautious on this. - when discussing Levy stable processes, a reference to a web page (Nolan) is not the correct citation. There are many works that can be cited, such as Samorodnitsky and Taqqu, Stable non-Gaussian random processes, 1994. - the authors mention FARIMA models, but these models are discrete. When discussing stochastic processes possessing scaling properties, no need to go to the discrete models. For an overview of the different stochastic models possessing such property, see e.g. Pipiras and Taqqu, Long-range dependence and self-similarity, 2017. - in the field of probability theory (and not finance, no need to go to these field, all was done in the field of mathematics) a Levy process is a continuous time stochastic processes having independent and stationary increments; it includes infinitely divisible families, and belongs to the general family of self-similar processes. - equation 3 is not correct, the good relation is beta=1+2H. For a Brownian motion, beta=2 and H=0.5.

The manuscript needs a complete rewriting from scratch with a much more simple and logical structure.

---

## Author Comment (AC2) · 25 Mar 2020

Dear Reviewer, thank you for taking time to review our manuscript. We would like to take the opportunity to discuss your arguments and to give some information which will also clarify and improve the manuscript.

1 - the authors should indicate clearly what is new in this manuscript with respect to these previous works Reply: We emphasize in line 32 (Âń This work discusses several statistical assumptions Âż . . .) that previous works has applied the FBm and the Levy Alpha stable distributions without or very few justifications about the underlining processes in the GNSS time series. For example, when can we use a family of distribution

such as the Levy family with infinite variance and heavy tails? What does that imply for the kind of stochastic processes defining the GNSS time series ? So far the literature on geodetic time series analysis is missing this discussion (see Line 37 " Therefore this work aims at understanding when the Levy processes can be applied to model geodetic time series. " ). Now, our methodology to investigate the use of these models, is based on the assumption of a third random variable to model the residual stochastic processes due to e.g. small transient signals, small jumps ( coseismic offsets ..) . . . This third random variable is defined a Levy process. The definition of this Levy process falls in three specific cases Levy Gaussian, Fractional Levy and stable Levy. We then develop a N-step method which is based on the estimation of the stochastic models when varying the length of the time series. 2- " the authors do not introduced adequately the topic: what is GNSS data, why does it have non-stationary and stochastic components? " Reply: Section 2 is here to describe step-by-step what we call a geodetic time series and the specificities of GNSS data. In Line "50", "GNSS time series are generally regarded as a sum of geophysical signals (i.e. seasonal signal, tectonic rate) and stochastic processes (. . .)". With our experience in previous publications, we have summarized the modelling and processing of GNSS time series. We discuss some of the fundamental hypothesis such as the Gauss-Markov assumption and the WSS hypothesis of the coloured noise. This coloured noise is a power-law noise (P(f) = 1/f^beta). In Section 2.2, we underline the relationship between this power-law noise, the fBm and the FARIMA. Therefore, the stochastic noise of the GNSS time series includes short and long memory processes. This topic is large and can be discussed comprehensively, but due to space limitation and clarity of the manuscript, we needed to restrain our introduction to GNSS time series, their stochastic properties and associated models. We refer to Montillet and Bos 2019, Chapter 2 for a longer discussion of the stochastic properties of the GNSS time series. Note that various references to previous works in mathematical geodesy are included, but the paper requires a minimum knowledge of geodetic time series to grasp this whole discussion. We will add some sentences at the beginning of Section 2 such as: "Geodetic time series consists out of

a set of observations at each epoch, containing noise which can be described as a set of multivariate random variables. The time series are modeled as a sum of a stochastic and functional (or trajectory) models. The functional model describes the geophysical processes intrinsic to the local and regional geodynamics of the area where the station is installed. The stochasticity originates from the unmodelled transient signals and the measurement errors (Williams et al., 2004; Bos et al., 2013; Montillet and Bos, 2019)."

3 – "the structure of the manuscript is very confusing, with many different models proposed from time to time, with no justification from the data." Reply: We think that there are two different issues in this comment. First, it seems that there is a confusion based on the definition of the GNSS time series. Therefore, we will clarify section 2 (including the few sentences written in the previous comment. Now, there are various stochastic noise models in GNSS time series analysis, because of the small geophysical signals (transients) and small offsets (due to local geodynamics or far away large magnitude Earthquake). We discuss that when formulating the assumptions about the Levy processes in Section 3.1 (see the assumptions behind Levy Gaussian, Fractional Levy and Stable Levy). In a nutshell, it has been formulated that the stochastic noise models of the GNSS time series are a sum of two random variables (r.v.) (also called stochastic processes as suggested by R1), modelling the white noise and coloured noise. The white noise is Gaussian distributed and the coloured noise is either a Flicker noise or a Power-law noise. Both are modeled via their covariance function as explained in Section 2 – eq. 2. Recently, it has been discussed the use of a third r.v. such as the random-walk (see discussions in He et al. 2017 and He et al. 2019). That is to model various transient signals (coseismic offsets, post-seismic relaxations . . .) which can or cannot be geophysically related. Note that one needs also to take into account the processing of GNSS time series which can generate outliers and spurious signals. However, the definition of this third r.v. is generally related to the local geophysical activity (e.g. postseimic events and small tremors generating a random-walk in stations located in Cascadia mountains – He et al. 2019, Montillet et al. 2018). Here, we propose to define this third r.v. using the family of Levy processes. The family of Levy

processes can model short and long memory processes and random jumps (Levy jump processes). However, it is not easy to model every time series with 3 r.v., because each time series is a unique sum of geophysical and stochastic processes. Therefore, we need to separate (as much as we can) the known geophysical signals (tectonic rate, seasonal signal) to the stochastic processes. Therefore, we have created this N-steps algorithm. By iterating the estimation of the stochastic and functional model, we can formulate the assumptions to characterize this 3 r.v. as defined in Table 1. In each step, we produce a residual time series which is the GNSS time series ÂÍminus" the estimated geophysical signals (e.g. tectonic rates and seasonal signal). Also between two steps, we increase the length of the time series. Note that the maximum increase in length is 1 year, because over a much longer time period we can introduce more small amplitude transient signals. For example, if there is no change (or negligible changes) in the estimated parameters of the stochastic noise models after N iterations, then the 3 r.v. is assumed to be a Gaussian Levy process following the properties of a pure Brownian motion. The processes are then assumed as short memory processes and are modeled as a Gaussian white noise. Thus, in this case we have a sum of two white Gaussian noises and a low-amplitude coloured noise. We postulate that the ARMA model should model the stochastic processes. The second case is when we have a noticeable change in the stochastic noise and functional models. That is when we have long-memory processes and high-amplitude coloured noise. The third r.v. is then chosen as a Fractional Levy process. In this case, the high-amplitude coloured noise produces long-memory processes and the FARIMA model should be used to model the stochastic processes. The last case is a special case when we have a large variance due to outliers or unmodelled signals of large amplitude, therefore there is anxiety in the chosen functional and stochastic models. We define the third r.v. as a stable Levy process which is directly related to the alpha stable Levy distribution.

Beyond that, there is the selection of the optimal noise model. This is another hot topic. Here, we have restricted the choice between the FL+WN and the PL+WN models. The choice is based on the maximization of the Akaike criterion. It is a pre-processing step

before the N-step algorithm. We justify our strategy based on the results in He et al. 2017 and He et al. 2019. A comprehensive discussion is in Montillet and Bos 2019 – Chapter 1 and 2. It will be confusing to integrate a full discussion on the optimal choice of the stochastic noise model, therefore for a matter of clarity we have restricted this part to line 220-225 (The optimal choice of the stochastic model . . .).

4 –"But no data is shown to justify this / Then show some plots of such data, with power spectrum and pdf. " Reply: Below (see Figure 1), we attach the GNSS time series of ASCO station, together with their power spectrum. These figures will be added in the annexes of the paper.

5- "explain how to estimate the parameters. " Reply: The stochastic noise model of the residual time series (ARMA, FARIMA, power-law + white noise . . . ) are estimated via maximum-likelihood using Hector software as discussed in Line 225-230. We have not emphasized the technique here, because it is also a long topic described in Montillet and Bos, 2019 ( see the first 6 chapters). The best estimator (maximum likelihood, Monte Carlo Markov Chain . . .) or the statistical "strategy" is out of the scope of this paper. Note that we have also discussed this point in the previous discussion during Review 1 (see point 4). In the final version of the manuscript, we will add a short appendix to discuss the estimation of the stochastic model jointly with the geophysical model using the log-likelihood.

6 – "when discussing Levy stable processes, a reference to a web page (Nolan) is not the correct citation. There are many works that can be cited, such as Samorodnitsky and Taqqu, Stable non-Gaussian random processes, 1994. " Reply: Thank you for this remark. We will add this reference in the final version.

7- "the authors mention FARIMA models, but these models are discrete. When discussing stochastic processes possessing scaling properties, no need to go to the discrete models. " Reply: The discrete models have been used in geodetic time series analysis (GNSS time series, tide gauges, see Chapter 2 in Montillet and Bos 2019). In

our present work, we use them to formulate assumptions on the stochastic noise properties of the GNSS time series (short vs long range dependencies, ..).The FARIMA model is interesting for our time series (and more widely for geodetic time series), because it has the ability to mode long-range dependencies due to the relation between the fractional index (d) and the Hurst parameter (H) (see Line 100-105). We discuss this ability over the less complex ARMA model in the case of high amplitude coloured noise (see discussion Line 135-140). Note that we will include this reference "Pipiras and Taqqu 2017"

8 – "equation 3 is not correct, the good relation is beta=1+2H. " Reply: It depends on how you define beta, following (https://www.ncbi.nlm.nih.gov/pmc/articles/PMC3947294/) If the definition is based on fractional Gaussian noise, beta =2H-1; if it is based on the fBm, it is based on beta =2H+1. Now, I use the definition from the geodetic community which is based on fractional Gaussian noise – see He et al. 2017 However, it is a good point to precise in the manuscript. We will add a sentence to indicate which definition we follow. Thank you.

[Figure]

**Fig. 1.** : ASCO time series (with functional model on top - red) for two stochastic noise models (PL+WN, FN+WN); Power spectrum (East (0), North (1), Up (2)); histogram of the residual time series.

---

## Editor Comment (EC1) · Stéphane Vannitsem (Editor) · 6 Apr 2020

Dear Dr Montillet

Thank you very much for your answers to reviewers' comments.

Both reviewers are rejecting the manuscript due to a considerable lack of clarity, logical structure and of an inappropriate description of the mathematical foundations. They both suggested to rewrite completely the manuscript. While evaluating the manuscript, I agree with both reviewers.

So my decision is to reject the paper in its present form. It will give you the opportunity to deeply revise your work and prepare a new version taking into account the constructive reviewers' comments.

You can also withdraw your manuscript. This will be notified to the reader of your manuscript that will stay online.

Thank you very much for considering our journal, Nonlinear Processes in Geophysics, for the publication of your work,

Stéphane Vannitsem, Handling editor